# Patient and health system delays in the diagnosis and treatment of tuberculosis in Gandaki, Nepal

**Bikram Singh Dhami**[1]*, **Damaru Prasad Paneru**[1], **Sagar Parajuli**[1], **K.C. Aarati**[1], **Dhirendra Nath**[1,2]

1 School of Health and Allied Sciences, Pokhara University, Pokhara, Gandaki Province, Nepal, 2 Health Directorate, Ministry of Social Development, Doti, Sudurpashchim Province, Nepal

* dhamibikram99@gmail.com

## Abstract

Delays in accessing healthcare worsen disease outcomes, increasing Tuberculosis (TB) transmission rates and mortality. Prolonged delays may contribute to drug-resistant TB strains in some cases, this study assessed delays in diagnosis and treatment among TB patients in Gandaki, Nepal. A cross-sectional study was conducted among a randomly selected sample of 194 TB patients enrolled in Direct Observed Treatment Short-course (DOTS) therapy. The data were collected through face-to-face interviews using a semi-structured interview schedule, which was developed through literature review and adaptation of the World Health Organization's multi-country study. Multivariate logistic regression was employed to identify factors associated with delays in diagnosis and treatment, considering a p value < 0.05 to indicate statistical significance. The median patient and health system delays were 35 (7–120) and 9 (2–98) days, respectively. Furthermore, 55.7% and 58.2% of patients experienced patient and health system delays, respectively. In the multivariable logistic regression analysis, factors associated with unacceptable patient delay included non enrollment in government health insurance programmes (AOR: 3.19; 95% CI: 1.29-7.98), seeking care from non-National Tuberculosis Program (non-NTP) providers (AOR: 3.19; 95% CI: 1.460-6.97), poor knowledge of TB (AOR: 3.74; 95% CI: 1.67-8.37), and high levels of perceived stigma (AOR: 3.15; 95% CI: 1.42-6.94). Furthermore, undergoing an initial diagnostic test other than GeneXpert (AOR: 3.25; 95% CI: 1.19-8.87) and visiting healthcare facilities multiple times before being diagnosed with TB (AOR: 5.62; 95% CI: 2.26-13.96) were significantly associated with unacceptable health system delay. Patient and health system delays were prevalent among TB patients. Reducing these delays is crucial for improving TB control. Therefore, urgent action is needed to implement education campaigns to improve TB literacy. Additionally, engaging private and informal healthcare providers and enhancing their capacity to deliver timely and effective TB care could potentially mitigate delays in diagnosis and treatment.

**Data availability statement:** All data are in the manuscript and Supporting Information files.

**Funding:** The authors received no specific funding for this work.

**Competing interests:** The authors have declared that no competing interests exist.

## Introduction

Tuberculosis (TB) is a major cause of illness and a leading cause of mortality and morbidity in low- and middle-income countries, accounting for 95% of new cases and 98% of tuberculosis-related deaths [1,2]. In 2021, tuberculosis caused 1.6 million deaths globally, making it the second leading infectious disease after COVID-19. Furthermore, the TB incidence rate has declined over the past two decades but increased by 3.6% in 2021 [3]. In 2014 and 2015, all member states of the WHO and the United Nations (UN) set a goal to end the TB epidemic by 2030, this commitment is being accelerated through the adoption of the WHO's End TB Strategy and the UN Sustainable Development Goals (SDGs) [4]. Six out of eight TB-burdened countries accounting for two-thirds of global TB cases are located in Asia: India, China, Indonesia, the Philippines, Pakistan and Bangladesh [5]. The South East Asia Region (SEAR) is a global hotspot of tuberculosis that contributes nearly half of all new TB cases worldwide and more than 50% of global TB deaths, excluding deaths from TB-HIV coinfection [3,6].

The rapid identification of TB patients depends on patients identifying symptoms early, seeking treatment and healthcare providers providing standard treatment services through the health system [7,8]. Delayed diagnosis and treatment of tuberculosis (TB) results in increased medication resistance, treatment failure, severe illness, increased mortality, and increased infectivity of *Mycobacterium tuberculosis* in the population [9]. Patients with undiagnosed TB act as reservoirs for transmission, and a delay in diagnosis results in the spread of TB, as one infection can lead to 10–25 secondary infections in a year. Ultimately, this can result in catastrophic health expenditures due to the severity of disease [10]. Despite improvements in tuberculosis (TB) knowledge and technological advancements, there remains a trend of missed TB cases within the healthcare system, resulting in an increased number of people at risk of TB transmission [11].

Nepal is a South Asian country with a high number of TB cases and is listed as a high burden country for MDR/RR-TB by the World Health Organization (WHO) [3]. The burden of TB in Nepal is marked by high incidence (245.1/100,000 population), prevalence (416.35/100,000 population) and mortality [12]. The adoption of the Stop TB strategy in 2006 and the End TB strategy in 2015 demonstrates the commitment of the government to end the TB epidemic in Nepal by 2035 [13]. Recent initiatives include digitalized case-based surveillance, preventive therapy for U5 children, public private mix (PPM), the TB-free *Palika* Initiative, the Electronic Tuberculosis (eTB) Register and FAST (Find Actively, Separating and Treating Effectively) [12,13]. The National Strategic Plan to End TB (2021/22–2025/26) has set the ambitious goal of reducing incidence and mortality rates, aiming to end the TB epidemic by 2035 and eliminate TB by 2050 [13]. The national TB program of Nepal fails to record more than half (58%) of TB cases every year, highlighting the limitations of diagnostic and surveillance processes [14].

The health system in Nepal includes both public and private providers offering tuberculosis diagnosis and treatment services. Public service providers include government hospitals, urban health clinics, and government health centers such as

primary health care centers (PHCCs) and health posts (HPs), while private providers encompass medical colleges, private hospitals and clinics [12]. A systematic review of studies conducted in both low- and high-income countries revealed a total delay of 25–185 days, patient delays of 4.9-162 days, and health system delays of 2–87 days on average [15]. Delays in TB diagnosis and treatment are associated with both patient and healthcare system characteristics [7,16,17]. Patient-related factors include the economic and sociodemographic characteristics of the patient, behavioral factors, awareness and knowledge of tuberculosis, stigma, and consultation with multiple health care providers (HCPs) [7,16,17]. The health system-related characteristics included physical inaccessibility to the facility, seeking care from private providers, and receiving diagnostic tests via GeneXpert [8,16,18].

Earlier evidence from Nepal also revealed a median total delay of 39.5 days, with a patient delay of 32 days and a health system delay of 3 days. The delays were due to both the system and patient characteristics [19]. TB ishighly stigmatized in Nepalese society and reported forms of stigma are perceived hatred and fear, discrimination by family and friends, reduced social interaction, blame and guilt for TB infection, institutional stigma and discrimination, and community discrimination [20]. However, their role in contributing to delays in TB diagnosis and treatment has not been well documented in the Nepalese context. This study bridges this gap by investigating how perceived stigma influences delays in diagnosis and treatment. Moreover, in Gandaki Province, the Case Notification Rate (CNR) of all forms of TB was 74 per 100,000 people in 2020/21 [14], indicating a potential delay in diagnosis. Similarly, three of the districts of Gandaki Province, Kaski, Gorkha and Nawalparasi, contributed 48% of the total notified TB cases in 2019/20 at the provincial level. The CNR of the Kaski district decreased from 92% to 78% in 2018/19 and slightly increased to 84% in 2019/20, which also reflects that there may be a delay in the diagnosis of TB [21]. Thus, this study aimed to assess the health seeking behaviour of TB patients and the factors contributing to delays in diagnosis and treatment among patients in the Kaski district of Gandaki Province, Nepal.

## Materials and methods

### Study setting

This study was conducted in Kaski District, the headquarters of Gandaki Province. The district comprises one Metropolitan City and four Rural Municipalities, with a total population of 600,051 distributed across 160,651 households, yielding an average household size of 3.74 persons [22]. The sex composition demonstrated a slight female predominance, with 48.9% males and 51.1% females. The population density within the district is 298 persons per square km. The multidimensional poverty index (MPI) of Gandaki Province was 0.035 in 2019, with a proportion of poor people of 9.6%. Additionally, the human development index (HDI) of Gandaki Province was 0.62, ranking second highest among the seven provinces of Nepal [23,24]. The Directly Observed Treatment, Short-course (DOTS) service is available at the Tuberculosis Treatment Center (TTC), two government hospitals, four PHCCs, 45 Health Posts (HPs), 18 urban health centers (UHCs) and three other health facilities in the Kaski district. Additionally, 21 private health facilities also provide DOTS services [25].

### Study design and study method

A cross-sectional study was conducted among 194 TB patients receiving treatment at DOTS centers in the Kaski district. Tuberculosis patients who underwent DOTS therapy and who had received treatment for a duration exceeding two weeks were the study participants. Participants aged 18 years or older were interviewed. The study excluded participants who were unable to properly provide information due to illness, as well as those who experienced treatment failure requiring retreatment or who were lost to follow-up.

### Sample size and selection

The sample size was determined using the formula for estimating the single population proportion [26]. Therefore, with a 53.21% delay of more than one month from a previous study published in Nepal [19], a margin error of 5%, 95% CI,

adjustments were made to account for a finite population (n = 325), and with a 10% nonresponse rate, the sample size was estimated to be 194. The required sample was selected as follows:

1. We selected 10 DOTS centers (S1 Table) with the highest patient loads (covering approximately 74% of the total patients) from 55 centers situated in the Kaski district, with at least one TB patient taking medication.

2. We allocated sample numbers (size) for each selected center using a probability proportionate to size (PPS) technique to ensure representation based on the center's patient load.

3. We then consecutively sampled individual patients from those attending the selected DOTS centers until we reached the required number of participants.

The sampling procedure involved the following steps:

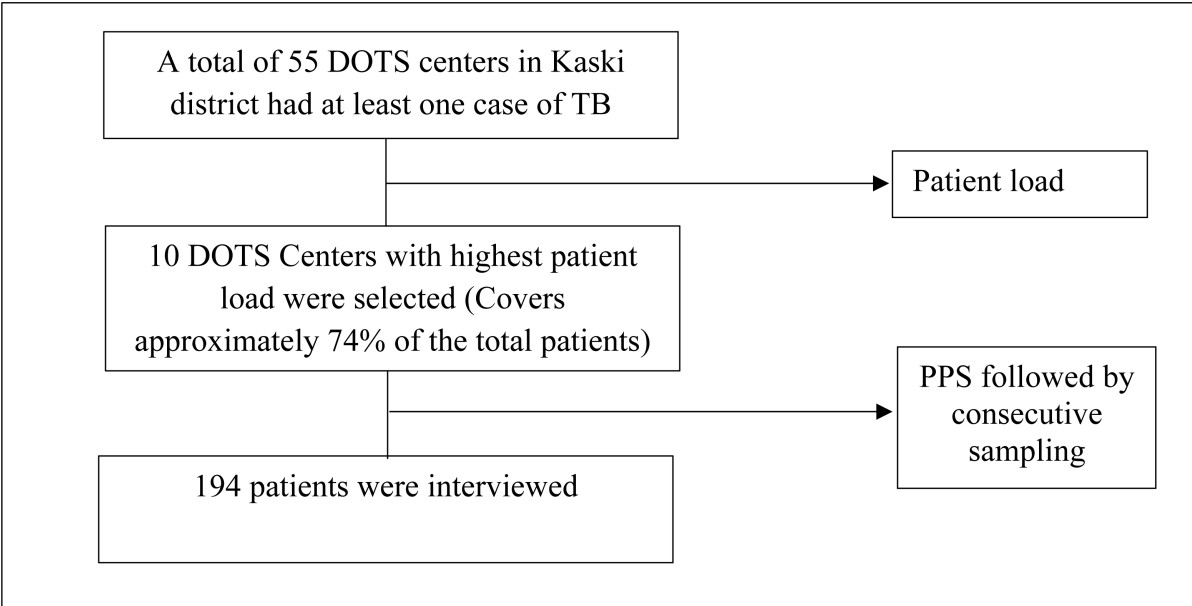

## Data collection techniques and tools

The data were collected using a face-to-face interview technique among selected participants using a pretested semi-structured interview schedule (S1 File) that was developed through literature review and adaptation of TB stigma and patient knowledge of TB from the World Health Organization's multi-country study on diagnosis and treatment delays in tuberculosis [7]. The researchers carefully reviewed and verified the questionnaire to ensure simplicity, clarity, and relevance to the research objectives. Furthermore, for ease of understanding, the questionnaire was back-translated (English-Nepali-English). The data were collected by two trained public health students using the Nepali language questionnaire during a one-month period from 15/01/2023–13/02/2023. The data collectors were oriented on study tools and data collection procedures before initiating field work.

## Variables

**Dependent variables.** Patient delay was assessed by asking the TB patients "In total, how long did you spend from the onset of illness to the first healthcare facility visit?", and the response time was recorded in days. If the obtained time was more than 30 days, the patient delay was considered unacceptable [19]. Likewise, health system delay was measured by asking "how long did it takes you to start TB treatment since you first visited health care facility?". If the

obtained time was more than 7 days, it was considered unacceptable health system delay [19,27]. The total delay was measured by subtracting the date of response to the question "When did the first symptoms of TB appear to you?" from the response to the question "When did you start anti-TB treatment?". A total delay greater than 28 days was considered unacceptable [19,28].

**Delays in diagnosis and treatment among tuberculosis patients:** Fig 1 illustrates the delays in diagnosis and treatment.

**Independent variables.  Health seeking behavior**: The health seeking behavior of the patients was assessed by asking the respondents about their first action after the onset of symptoms, which facility they had visited first, the reason for visiting such a facility and their perceived reason for delayed consultation with formal service providers.

**Clinical characteristics**: Clinical characteristics of the patients were type of TB present (pulmonary vs extra pulmonary), presence of HIV comorbidity, presence of other chronic disease comorbidities, type of health facility at which the final diagnosis was made, history of contact with TB-infected people in the last year and place of contact and smear status (positive and negative).

**TB-related knowledge:** Study participants' knowledge of TB was assessed using seven questions on knowledge about TB, curability, contagiousness, the availability of TB vaccines, and the approximate duration of treatment. Knowledge of tuberculosis was measured on a 3-point Likert scale (0 for highest knowledge, 2 for lowest knowledge) [7]. After reversing the score, the total knowledge score was computed. Respondents who scored above the median were labelled "good knowledge", while those with scores less than or equal to the median score were labelled "poor knowledge".

**Perceived stigma:** Stigma associated with TB among the study participants was assessed using a five-point Likert scale, and 14 negative statements were used to assess perceived TB stigma. The questions included shame associated with TB, having to hide tuberculosis diagnosis from others, the cost incurred by the long disease duration, isolation due to tuberculosis, girls' autonomy in deciding about receiving tuberculosis treatment, and the extent to which tuberculosis affects the following: relations with others, work performance, marital relations, family responsibility, chances of marriage, family relations, female infertility, complications during pregnancy, breastfeeding and pregnancy outcomes. The variable measuring stigma was recorded on a 5-point Likert scale (0, the highest, and 4, the lowest degree of stigma) [7]. The

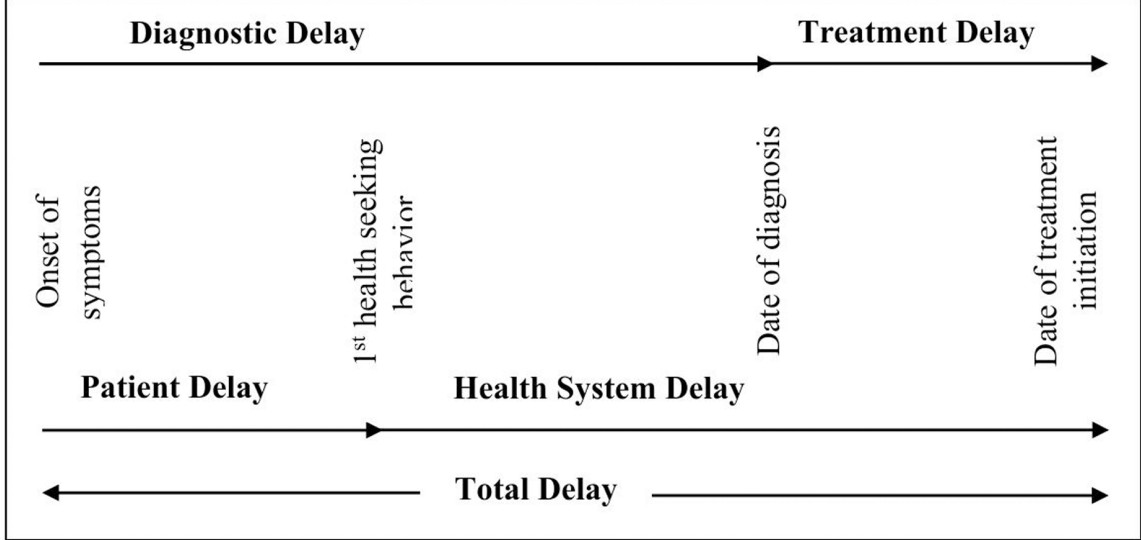

**Fig 1.  Delays in diagnosis and treatment among tuberculosis patients.**

score was first reversed, computed, and dichotomized into "High Stigma" and "Low Stigma" categories considering the median score as the cut-off.

**Control variables.** The control variables were age (in years), family type, religion, ethnicity, education, occupation, enrollment in the health insurance program, distance to the health facility, smoking status, alcohol use and body mass index (BMI).

Ethnic groups were classified according to the Health Management Information System (HMIS) of Nepal, which includes Dalit, Disadvantaged Janjati, Disadvantaged Non-Dalit Terai Cast, Religious Minorities, Relatively Advantaged Janjati, and Upper Cast Group. Religion was recorded as Hindu, Buddhist, Muslim or Christian. Family type was identified as nuclear, joint and extended. Those who could not read and write in Nepali, recorded as illiterate and literate people, were further classified according to their highest education level into "Grade 1-8", "Grade 9-12" and university level. The enrollment status in the National Health Insurance Program was dichotomized as "yes" or "no". Furthermore, the distance to the nearest health facility was captured on a continuous scale but later categorized into "≤30 minutes" and ">30 minutes". Those who used to smoke earlier but who were not currently smoking were categorized as "Quitted Smoking", those who had never smoked were categorized as "Never", and currently smoking respondents were classified as "Current Smokers". Likewise, those who used to drink earlier but did not drink currently were classified as "Past users", those who never drink in their lifetime were categorized as "Never", and those who currently consume alcohol were identified as "Current users". BMI was calculated using patients weight and height. Furthermore, the obtained result was grouped into four categories, "underweight", "normal", "overweight", and "obese", as classified by the World Health Organization (WHO) [29].

## Quality assurance

Pretesting was conducted among 20 TB patients recruited from Matrisishu Miteri Hospital and Lekhnath Health Post in the Kaski District. The pretested data were not included in the final analysis. After pretesting, the sequence of the questions was rearranged, and few questions were found to be difficult for the participants to understand and were rephrased. During data collection, the questionnaire was checked immediately after each interview for completeness. Cronbach's alpha was used to measure the internal consistency of the survey instrument. The Cronbach's alpha for knowledge was 0.63, and it was 0.83 for stigma. The HMIS 6.4 A (TB treatment card-health facility) and HMIS 6.4 B (TB treatment card-patient) were cross-checked to ensure the quality of the data. The HMIS 6.4A and HMIS 6.4B are treatment cards used to record the personal and disease-related test results of the TB patient, details of the patient's daily medication intake, treatment results, etc., in the health management information system.

## Data management and statistical analysis

The collected data were entered into EpiData version 3.1, where checks were implemented to prevent human error during data entry. However, EpiData have limitations in terms of error prevention, and a manual check was conducted on 10% of the data. The verified data were exported to SPSS (Statistical Package of Social Sciences) version 21 for further analysis. Categorical data were examined in terms of numbers and percentages, while continuous data were described using metrics such as the mean, standard deviation, median, and IQR (minimum-maximum). The multivariable binary logistic regression was used to identify any relationships between patient delay and health system delay with socio-demographic and behavioral characteristics, health-seeking behavior, clinical characteristics, TB Knowledge, and TB stigma. The multivariable logistic regression model incorporated all statistically significant ($p \leq 0.05$) independent variables identified in the binary logistic regression analysis.

Multi collinearity among the independent variables was assessed using variance inflation factor (VIF) analysis. The VIFs for the independent variables were less than two, with the highest VIF observed being 1.88 for the variable "Education status". We used the Hosmer–Lemeshow chi-square test to determine the goodness of fit of the logistic regression model, and the results revealed that the model was a good fit, with $p > 0.05$. To identify unacceptable delays in

tuberculosis (TB) diagnosis, the sample was dichotomized using cut-off points of 30 days for patients and 7 days for the health system [19,27,28]. Thirty days as the cut-off for patients was used because patients who remained untreated for one month since showing clinical signs have a negative impact on clinical outcome [30,31]. However, the decision to use 7 days as a cut-off for the health system was based on previous studies published in Nepal [19,27].

### Ethical consideration

This study obtained ethical approval from the Pokhara University Institutional Review Committee (IRC), with reference number 78–079/80. Permission to conduct the study was granted by the Health Office in the Kaski district. Respondents were informed about the study purpose, and written informed consent was obtained prior to conducting the interviews. The interviews were conducted in the DOTS room of a selected health facility to ensure privacy and confidentiality. Instead of using participants' name, numerical coding was used as an identifier. Participation in the interviews was voluntary, and participants were informed that they could decline to respond to specific questions at any time during the interview process or discontinue the whole interview.

## Results

### Sociodemographic characteristics of the study participants

Among the 194 participants, 62.0% were male, and majority (76.25%) were Hindu. A total of 36.6% of the participants belonged to the upper cast group, followed by the Relatively Advantaged Janjatis group (23.7%). Education attainment varied; around a third (35.6%) had completed Grade 9–12, while almost two-thirds (32.47%) had never received formal education. Of the total, 17% were employed as daily wage workers, and 15.4% were students. Approximately one-fourth of the participants (23.7%) were enrolled in a government health insurance scheme. Furthermore, three-fifth (60.30%) of the participants reported ever having consumed tobacco products, while just more than half (52.57%) reported drinking alcohol at some point in their lives. Moreover, approximately one-third (32%) of the participants were classified as underweight (Table 1).

### Clinical characteristics and health-seeking behavior

The cough was major symptom experienced by more than half (59.3%), followed by fever (38.1%) and weight loss (36.1%). Chest pain caused 25.3% of the patients to seek healthcare. Over half of the participants (51.3%) were diagnosed with pulmonary tuberculosis, while 39.7% presented with extra pulmonary tuberculosis. Half of the participants showed a smear-positive status, the majority (88.1%) tested negative for HIV, and 9.8% had an unknown HIV status. Additionally, more than one-fourth of the participants reported the presence of comorbidities other than HIV (Table 2).

Furthermore, just over half (54.1%) of the participants sought medical care at modern healthcare facilities. Additionally, 26.8% of the participants reported that they had to travel for a duration of 30 minutes or more to reach the healthcare facility. Among modern medical facilities, private facilities were found to be the most frequently chosen. A notable proportion of the participants (69.6%) had visited two or more healthcare facilities before receiving a final diagnosis, while two-thirds (66.5%) had made more than three visits to healthcare facilities. Furthermore, only few (15.5%) reported contact with TB patients in past year (Table 3).

### Tuberculosis treatment-related characteristics

A majority of the respondents (58.2%) experienced health system delays. Furthermore, more than half of the participants (52.06%) started prompt treatment after being diagnosed with tuberculosis. Moreover, an equal proportion of participants (26.88%) identified distant residence and the absence of DOTS provider as the primary reasons for not initiating treatment promptly (Table 4).

**Table 1. Sociodemographic and lifestyle-related characteristics of tuberculosis patients in Kaski, District (n = 194).**

| Variables | Frequency (n) | Percentage (%) |
|---|---|---|
| **Age** | | |
| 18-35 years | 93 | 47.9 |
| ≥ 35 years | 101 | 52.1 |
| Median (IQR): 37(18–94) years | | |
| **Sex** | | |
| Male | 120 | 62.0 |
| Female | 74 | 38.0 |
| **Type of Family** | | |
| Nuclear | 110 | 56.7 |
| Joint/Extended | 84 | 46.3 |
| **Religion** | | |
| Hindu | 149 | 76.8 |
| Buddhist | 36 | 18.6 |
| Muslim | 4 | 2.1 |
| Christian | 5 | 2.6 |
| **Ethnicity** | | |
| Dalit | 27 | 13.9 |
| Disadvantaged Janjati | 39 | 20.1 |
| Disadvantaged Non-Dalit Terai Cast | 7 | 3.6 |
| Religious Minorities | 4 | 2.1 |
| Relatively Advantaged Janjati | 46 | 23.7 |
| Upper Cast Group | 71 | 36.6 |
| **Marital Status** | | |
| Currently Married | 115 | 59.3 |
| Divorced/Separated | 2 | 1.0 |
| Widow/Wider | 28 | 14.4 |
| Never Married | 49 | 25.3 |
| **Educational Status** | | |
| Illiterate | 38 | 19.6 |
| Informal Education | 25 | 12.9 |
| Grade 1–8 | 44 | 22.7 |
| Grade 9–12 | 69 | 35.6 |
| University Degree | 18 | 9.3 |
| **Occupation** | | |
| Agriculture | 17 | 8.8 |
| Business | 26 | 13.4 |
| Services | 19 | 9.8 |
| House maker | 21 | 10.8 |
| Daily wage worker | 33 | 17.0 |
| Unemployed | 11 | 5.7 |
| Student | 29 | 15.4 |
| Dependent population | 23 | 11.8 |
| Others | 15 | 7.7 |
| **Health Insurance Enrolment Status** | | |
| No | 148 | 76.3 |
| Yes | 46 | 23.7 |

*(Continued)*

**Table 1.** (Continued)

| Variables | Frequency (n) | Percentage (%) |
|---|---|---|
| **Smoking status** | | |
| Never | 77 | 39.7 |
| Current smoker | 48 | 24.7 |
| Quitted smoking | 69 | 35.6 |
| **Alcohol use** | | |
| Never | 92 | 47.4 |
| Current user | 22 | 11.3 |
| Past user | 80 | 41.3 |
| **BMI** | | |
| Underweight | 62 | 32.0 |
| Normal | 101 | 52.1 |
| Overweight | 23 | 11.9 |
| Obese | 8 | 4.1 |

IQR=Inter Quartile Range

### Knowledge and perceived stigma related to tuberculosis

The majority of respondents (80%) had prior knowledge of tuberculosis before being diagnosed with TB. More than half of the participants (57.2%) had poor knowledge of TB, while nearly half of the respondents (45.4%) experienced a high degree of stigma associated with TB (Table 5).

### Delays in Diagnosis and Treatment among Tuberculosis Patients in Kaski, District

Patients took an average of 38±23 days and a median of 35 (IQR: 24–45) days to first visit a health care service provider after the onset of their symptoms. Similarly, the mean and median number of days the health system delayed from first contact by patients with HCFs to the initiation of treatment were 12±13 days and 9 (IQR: 5–13) days, respectively (Fig 2).

### Factors associated with patient delay

According to the multivariable logistic regression, non enrollment in government health insurance (AOR: 3.19; 95% CI: 1.29-7.98), seeking care from non-NTP providers (AOR: 3.19; 95% CI: 1.46-6.97), having poor knowledge of TB (AOR: 3.74; 95% CI: 1.67-8.37), and experiencing high levels of perceived stigma (AOR: 3.15; 95% CI: 1.42-6.94) were independently associated with greater odds of patient delay beyond the median of 30 days (Table 6).

### Factors associated with health system delay among tuberculosis patients in Kaski district

Initial diagnostic tests other than GeneXpert (AOR: 3.25; 95% CI: 1.19-8.87) and visiting healthcare facilities three or more times before being diagnosed with TB (AOR: 5.62; 95% CI: 2.26-13.96) were significantly associated with greater odds of unacceptable health system delay (Table 7).

### Factors associated with total delay among tuberculosis patients in the Kaski district

Seeking care from non-NTP providers (AOR: 5.19; 95% CI: 1.53-17.61), having poor knowledge of TB (AOR: 5.40: 95% CI: 1.60-18.21), having high perceived TB stigma (AOR: 6.52: 95% CI: 1.54-27.57), and seeking care from two or more health care facilities (AOR: 3.86: 95% CI: 1.19-12.45) were significantly associated with greater odds of unacceptable total delay (Table 8).

**Table 2. Clinical behaviour related characteristics of tuberculosis patients in the Kaski district (n = 194).**

| Variables | Frequency (n) | Percentage (%) |
|---|---|---|
| **Clinical sign and symptoms (n = 194)*** | | |
| Cough | 115 | 59.3 |
| Fever | 74 | 38.1 |
| Weight loss | 70 | 36.1 |
| Hemoptysis | 21 | 10.8 |
| Chest pain | 68 | 35.1 |
| Night sweating | 19 | 9.8 |
| Breathlessness | 69 | 35.6 |
| Loss of appetite | 60 | 30.9 |
| Pleural fluid | 22 | 11.3 |
| Enlargement of gland | 21 | 10.8 |
| Others[a] | 35 | 18.04 |
| **Symptoms that made to seek care (n = 194)*** | | |
| Cough | 22 | 11.3 |
| Fever | 15 | 7.7 |
| Weight loss | 13 | 6.7 |
| Hemoptysis | 17 | 8.8 |
| Chest pain | 49 | 25.3 |
| Breathlessness | 31 | 16.0 |
| Enlargement of gland | 21 | 10.8 |
| Severe back pain | 7 | 3.6 |
| Others[a] | 19 | 9.8 |
| **Type of Tuberculosis** | | |
| PBC | 95 | 49.0 |
| PCD | 22 | 11.3 |
| EPTB | 77 | 39.7 |
| **Sputum examination result** | | |
| Positive | 99 | 51.0 |
| Negative | 95 | 48.5 |
| Not done | 1 | 0.5 |
| **HIV status** | | |
| Positive | 4 | 2.1 |
| Negative | 171 | 88.1 |
| Unknown | 19 | 9.8 |
| **Presence of other chronic diseases[b]** | | |
| Yes | 54 | 27.8 |
| No | 140 | 72.2 |
| **Diagnostic Test** | | |
| Microscopy and Chest X-ray | 37 | 19.07 |
| GeneXpert | 58 | 29.90 |
| FNAC | 19 | 9.79 |
| Pleural biopsy | 31 | 15.98 |
| CT-Scan | 25 | 12.88 |
| Others[c] | 24 | 12.37 |

[a]Sweating, severe headache, stomach pain, fatigue,

[b]Diabetes, Arthritis, Epilepsy, COPD, Chronic Kidney Disease, HTN, Heart Problem, PBC, Pulmonary Bacteriologically, PCD, Pulmonary Clinically Diagnosed, EPTB, Extra-Pulmonary Tuberculosis,

[c]Endoscopy, Mantoux test, Culture, MRI

**Table 3. Health seeking behavior among Tuberculosis patients in Kaski, District (n = 194).**

| | | |
|---|---|---|
| **The first action to the current illness (n = 194)** | | |
| Self-Medication | 22 | 11.3 |
| Visited Pharmacy/Clinic | 56 | 28.9 |
| Traditional healer | 11 | 5.7 |
| Visited a modern health facility | 105 | 54.1 |
| **Type of Health facility visited (n = 105)** | | |
| UHC/Health post | 5 | 4.8 |
| Government Hospital | 32 | 30.5 |
| Private Hospital | 61 | 58.1 |
| Tuberculosis Treatment Center | 7 | 6.7 |
| **Reasons for the first consultation with a modern health facility (n = 105)*** | | |
| Accessible | 101 | 96.2 |
| Confident getting cured | 74 | 70.5 |
| Free services | 27 | 25.7 |
| Advised by somebody | 10 | 9.5 |
| Referred by previous services | 33 | 31.4 |
| Others | 3 | 2.8 |
| **Reasons for non-consultation with health facility (n = 89)*** | | |
| Too far | 5 | 5.6 |
| Too busy/long waiting time | 39 | 43.8 |
| Bad experience | 8 | 9.0 |
| Thought symptoms are not serious | 89 | 91.0 |
| Others | 12 | 13.5 |
| **Time to reach the health facility of first contact (n = 194)** | | |
| < 30 min | 142 | 73.2 |
| ≥ 30 min | 52 | 26.8 |
| **Time spent from the onset of symptoms to the first HCF visit (n = 194)** | | |
| ≤ 30 days | 86 | 44.3 |
| > 30 days | 108 | 55.7 |
| **Thought delayed consultation (n = 194)** | | |
| Yes | 128 | 66.0 |
| No | 66 | 34.0 |
| **Reasons for Delayed consultation (N = 128)*** | | |
| Hoped symptoms would go away by themselves | 117 | 91.4 |
| Lack of Money to cover consultation fees | 36 | 28.1 |
| Busy occupational life | 62 | 48.4 |
| Transport and long distance to HCF | 22 | 17.2 |
| Bad staff attitude to patient | 9 | 7.0 |
| Others | 19 | 13.3 |
| **HCF Visited (n = 194)** | | |
| 1 | 59 | 30.4 |
| 2 or more | 135 | 69.6 |
| **Number of visits to HCF (n = 194)** | | |
| ≤ 3 visits | 65 | 33.5 |
| > 3 visits | 129 | 66.5 |
| **Time spent from first HCF visit to Final Diagnosis** | | |
| ≤ 7 days | 81 | 41.8 |

*(Continued)*

**Table 3.** (Continued)

| | | |
|---|---|---|
| **The first action to the current illness (n = 194)** | | |
| > 7 days | 113 | 58.2 |
| **Thought Delayed Diagnosis (n = 194)** | | |
| Yes | 104 | 53.6 |
| No | 90 | 46.4 |
| **Perceived reason for delayed diagnosis (n = 104)[*]** | | |
| Failure of providers to diagnose | 80 | 76.92 |
| Prescription of unnecessary drugs | 50 | 48.07 |
| Repeated referral to facilities | 13 | 12.50 |
| Other | 8 | 7.69 |
| **Health care facility that made the final diagnosis (n = 194)** | | |
| Public Hospital | 52 | 26.8 |
| Private Hospital | 76 | 39.2 |
| Tuberculosis Treatment Centre | 66 | 34.0 |
| **History of TB contact within the last one year (n = 194)** | | |
| Yes | 30 | 15.5 |
| No | 164 | 84.5 |
| **Point of Contact (n = 30)** | | |
| Household | 13 | 43.3 |
| Workplace | 12 | 40.0 |
| School/College | 5 | 16.7 |

[*]Multiple responses

**Table 4. Tuberculosis treatment-related characteristics of tuberculosis patients in Kaski, district (n = 194).**

| Variables | Frequency (n) | Percentage (%) |
|---|---|---|
| **Time spent initiating treatment since the first HCF visit (n = 194)** | | |
| ≤ 7 days | 81 | 41.8 |
| > 7 days | 113 | 58.2 |
| **Initiation of Treatment after Diagnosis (n = 194)** | | |
| Immediately | 101 | 52.06 |
| Taken time | 93 | 47.94 |
| **Reasons for not initiating treatment immediately (n = 93)[*]** | | |
| Reluctant to initiate the treatment | 22 | 23.65 |
| Fear of long treatment | 18 | 19.35 |
| Absence of DOTS provider | 25 | 26.88 |
| Residence is far from HCF | 25 | 26.88 |
| Too ill to initiate treatment | 20 | 21.50 |

[*]Multiple response

## Discussion

This study aimed to identify delays in TB diagnosis and treatment and associated factors among TB patients in Kaski, Nepal. The median (IQR) patient and health system delays were 35 (7–120) days and 9 (2–98) days, respectively. The major findings were as follows: a) statistically significant factors associated with greater odds of patient delay beyond

**Table 5. Knowledge and perceived stigma related to tuberculosis patients in Kaski, District (n = 194).**

| Variables | Frequency (n) | Percentage (%) |
|---|---|---|
| **Heard about TB before diagnosed with TB (n = 194)** | | |
| Yes | 155 | 79.9 |
| No | 39 | 20.1 |
| **Source of information (n = 155)** | | |
| Media | 24 | 15.5 |
| Healthcare facility/HCP | 28 | 18.1 |
| By studying | 61 | 39.4 |
| Friends/Neighbor | 42 | 27.1 |
| **TB-related Knowledge (n = 194)** | | |
| Good | 83 | 42.8 |
| Poor | 111 | 57.2 |
| **Perceived Stigma (n = 194)** | | |
| High | 88 | 45.4 |
| Low | 106 | 54.6 |

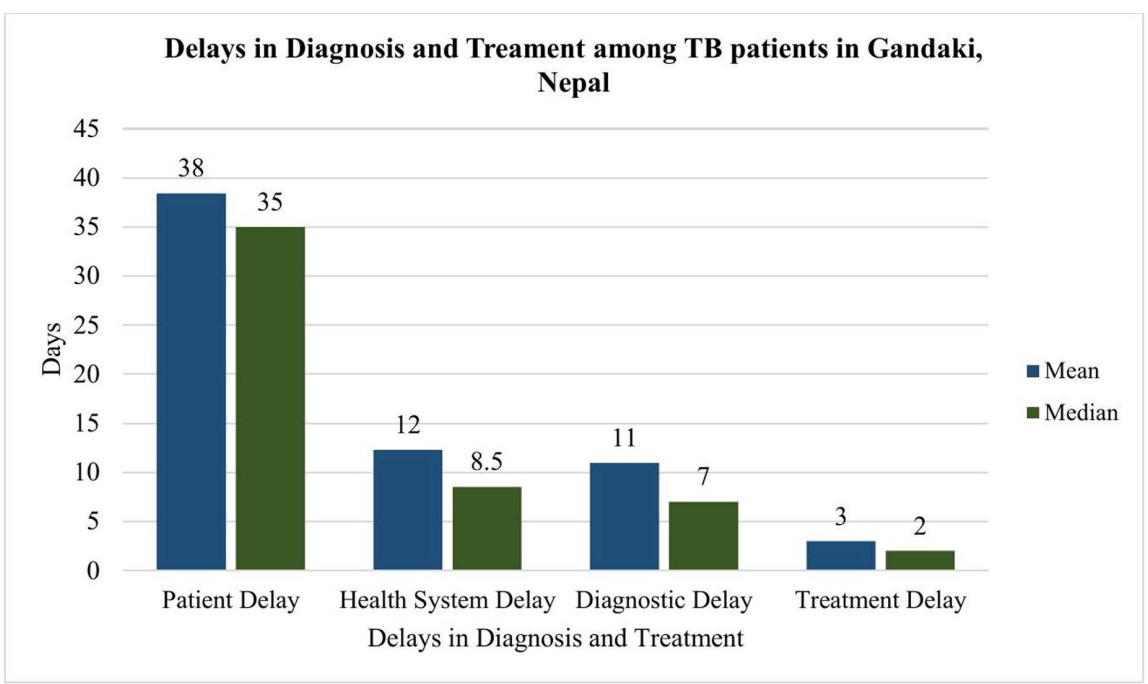

**Fig 2. Delays in diagnosis and treatment among tuberculosis patients in Gandaki, Nepal.**

the median of 30 days, including, non-enrollment in government health insurance programmes, seeking care from non-NTP providers, poor knowledge of TB, and high levels of perceived stigma; b) statistically significant factors for unacceptable health system delay, including initial diagnostic tests other than GeneXpert and visitinghealth facilities multiple times before being diagnosed with TB; and c) statistically significant factors for unacceptable total delay, including poor knowledge of TB, having high perceived TB stigma, seeking care from multiple healthcare facilities prior to the final diagnosis of TB and initially visiting non-NTP providers after the onset of symptoms.

**Table 6. Factors associated with patient delay among tuberculosis patients in the Kaski district (n = 194).**

| Variables | Frequency (n) | Unadjusted | | Adjusted | |
|---|---|---|---|---|---|
| | | OR (95%CI) | *P value* | OR (95%CI) | *P value* |
| **Age** | | | | | |
| 18-40 years | 109 | Ref. | 0.053 | | |
| > 40 years | 85 | 1.77 (0.99–3.17) | | | |
| **Marital status** | | | | | |
| Married | 145 | **2.50 (1.29–4.88)*** | 0.007 | 1.60 (0.64–4.02) | 0.315 |
| Unmarried | 49 | Ref. | | Ref. | |
| **Education status** | | | | | |
| Formal | 131 | Ref. | <0.001 | Ref. | 0.067 |
| Informal | 63 | **4.84 (2.40–9.76)*** | | 2.59 (0.94–7.20) | |
| **Occupation** | | | | | |
| Daily Wage Worker | 33 | **2.44 (1.07–5.58)*** | 0.034 | 1.43 (0.48–4.20) | 0.521 |
| Othersa | 161 | Ref. | | Ref. | |
| **Enrollment in health insurance program** | | | | | |
| Yes | 46 | Ref. | <0.001 | Ref. | **0.012** |
| No | 148 | **3.50 (1.73–7.04)** | | **3.19 (1.29–7.98)*** | |
| **Initially, visited health facility** | | | | | |
| Non-NTP provider | 89 | **3.55 (1.94–6.48)*** | <0.001 | **3.19 (1.46–6.97)*** | **0.004** |
| NTP Provider | 105 | Ref. | | Ref. | |
| **Walking Distance to HCF of first contact** | | | | | |
| > 30 Minutes | 52 | **2.79 (1.39–5.60)** | <0.004 | 1.42 (0.58–3.48) | 0.441 |
| ≤ 30 Minutes | 142 | Ref. | | Ref. | |
| **Smear status** | | | | | |
| Positive | 99 | **2.12 (1.19–3.77)*** | 0.011 | 1.36 (0.64–2.89) | 0.423 |
| Negative | 96 | Ref. | | Ref. | |
| **Smoking status** | | | | | |
| No | 77 | Ref. | 0.021 | Ref. | 0.290 |
| Yes | 117 | **1.99 (1.11–3.57)*** | | 0.55 (0.18–1.66) | |
| **TB-related Knowledge** | | | | | |
| Poor | 111 | **8.53 (4.45–16.36)*** | <0.001 | **3.74 (1.67–8.37)*** | **0.001** |
| Good | 83 | Ref. | | **Ref.** | |
| **Perceived Stigma** | | | | | |
| Low | 106 | Ref. | <0.001 | Ref. | **0.004** |
| High | 88 | **6.24 (3.28–11.87)*** | | **3.15 (1.42–6.94)*** | |

*Statistically significant at p < 0.05, Ref. = reference value,

aAgriculture, Services, Business, House maker, students, and unemployed

### Factors associated with patient delay

This study demonstrated a long delay of TB diagnosis in Gandaki Province, with a median delay of 35 days. This is far more than the ideal time for TB diagnosis within 14–21 days after the onset of the first symptom [32]. These results are comparable to those of studies conducted in Ethiopia [28,33], which reported a median patient delay of 30 days, but greater than those of other studies, which reported median patient delays of 18 (IQR: 8–34) [34] and 17 days (IQR: 9–33) [35].

In our study it was observed that patients who sought care from non-NTP providers were three times more likely to experience delays compared to those who consulted NTP providers. Despite this more than 45% patient sought care from

**Table 7. Factors associated with health system delay among tuberculosis patients in the Kaski district (n = 194).**

| Variables | Frequency (n) | Unadjusted | | Adjusted | |
|---|---|---|---|---|---|
| | | OR (95%CI) | *P value* | OR (95%CI) | *P value* |
| **Age** | | | | | |
| 18-40 years | 109 | Ref. | 0.465 | | |
| > 40 years | 85 | 1.24 (0.70–2.21) | | | |
| **Marital Status** | | | | | |
| Married | 145 | 1.33 (0.69–2.55) | 0.395 | | |
| Unmarried | 49 | Ref. | | | |
| **Education Status** | | | | | |
| Formal | 131 | Ref. | 0.305 | | |
| Informal | 63 | 1.38 (0.75–2.56) | | | |
| **Occupation** | | | | | |
| Daily wage worker | 33 | **0.40 (0.18–0.85)***  | 0.018 | 0.55 (0.21–1.44) | 0.222 |
| Others[a] | 161 | Ref. | | Ref. | |
| **Enrollment in Health Insurance Program** | | | | | |
| Yes | 46 | Ref. | 0.451 | – | |
| No | 148 | 0.77 (0.39–1.52) | | | |
| **Smear status** | | | | | |
| Positive | 99 | Ref. | <0.001 | Ref. | 0.328 |
| Negative | 106 | **5.42 (2.89–10.18)*** | | 1.72 (0.58–5.06) | |
| **Type of Tuberculosis** | | | | | |
| Extra Pulmonary | 77 | **4.77 (2.46–9.22)*** | <0.001 | 1.02 (0.35–3.04) | 0.965 |
| Pulmonary | 117 | Ref. | | Ref. | |
| **Initially, visited health facility** | | | | | |
| Non-NTP providers | 89 | **1.80 (1.01–3.20)*** | 0.047 | 1.93 (0.92–4.06) | 0.084 |
| NTP providers | 105 | Ref. | | Ref. | |
| **No. of HCF Visited** | | | | | |
| 1 HCF | 59 | Ref. | <0.001 | Ref. | 0.240 |
| ≥ 2 HCFs | 135 | **3.48 ( (1.84–6.59)*** | | 1.74 (0.69–4.40) | |
| **No. of Times HCF visited** | | | | | |
| <3 Visits | 65 | Ref. | <0.001 | Ref. | **<0.001** |
| ≥ 3 visits | 129 | **6.50 (3.36–12.59)*** | | **5.62 (2.26–13.96)*** | |
| **HCF that made final diagnosis** | | | | | |
| Government institution | 52 | **3.14 (1.56–6.32)*** | 0.001 | 1.87 (0.76–4.58) | 0.171 |
| Private institution | 76 | 1.40 (0.68–2.90) | 0.366 | 1.58 (0.64–3.93) | 0.324 |
| Tuberculosis Treatment Center | 66 | Ref. | | Ref. | |
| **Initial diagnostic test used** | | | | | |
| Others[b] | 136 | **5.15 (2.65–10.02)*** | <0.001 | **3.25 (1.19–8.87)*** | **0.022** |
| Gene Xpert | 58 | Ref. | | Ref. | |
| **Contact with TB patient within one year** | | | | | |
| No | 30 | **2.84 (1.27–6.37)*** | 0.011 | 1.80 (0.67–4.85) | 0.245 |
| Yes | 164 | Ref. | | Ref. | |
| **Tuberculosis related Knowledge** | | | | | |
| Poor | 111 | 1.23 (0.69–2.18) | 0.490 | – | |
| Good | 83 | Ref. | | | |

*(Continued)*

**Table 7.** (Continued)

| Variables | Frequency (n) | Unadjusted | | Adjusted | |
|---|---|---|---|---|---|
| | | OR (95%CI) | *P value* | OR (95%CI) | *P value* |
| **Perceived Stigma** | | | | | |
| Low | 106 | Ref. | 0.125 | – | |
| High | 88 | 0.64 (0.36–1.13) | | | |

*Statistically significant at p < 0.05, Ref. = Reference value, HCF = Healthcare Facility,

ªAgriculture, Services, Business, House maker, students, and unemployed,

ᵇMicroscopy and chest X-ray, Culture, FNAC, Pleural biopsy and CT scan

private health care providers, similar experience was documented in studies conducted in India [36,37]. Most of patients in South Asia reflect similar health seeking behavior as private sector providers are more accessible and available [38,39], and ensure confidentiality that offers protection against stigma [37]. Poor knowledge of TB was significantly associated with a delay in TB diagnosis. These findings were also consistent with those of previous studies. A systematic review of studies conducted in the Middle East and North Africa revealed that poor knowledge regarding TB was associated with patient delays [40]. Furthermore, patient misconceptions about curability and perceptions of DOTS services were found to be independent predictors of TB diagnosis [41]. This is because patients who believe that they have TB because of some evil forces and do not consider the symptoms to be severe fear social stigmatization often try to hide their illness and rely more on self-medication and other alternative forms of treatment, leading to treatment delays. The findings imply that adequate and innovative awareness-raising activities should be conducted by concerned bodies to increase knowledge regarding TB among the general population. A study by Dakito et al. in Ethiopia revealed that lack of awareness about the severity of symptoms, resorting to other alternative providers such as spiritual healers and drug vendors, poverty, high stigma and the number and types of facilities visited were the primary factors for delaying health service care by TB patients [42].

In a country such as Nepal, where poverty is still a major problem, people with TB face several socioeconomic barriers to TB diagnosis and treatment, such as food insecurity, high travel and food costs and loss in income [43]. High levels of stigma further exacerbate this situation. Like for leprosy and HIV, TB diagnosis and being seen as a person taking medicine can result in a decline in social participation, perceived stigma and concealment of diseases among the larger community, resulting in a delay in treatment [43–45]. In addition, mistreatment by family members and a culture of blame and shame, especially by higher caste families, are still prevalent in Nepalese communities, leading to a lack of treatment adherence [43]. Interventions such as peer support programs to serve as a source of guidance and motivation for another person facing stigma regarding TB and collaboration between NGOs, the public sector and advocacy groups working on health and human rights to leverage resources and expertise to end the stigma and discrimination against TB might be useful.

Similarly, when asked about the major reason behind delayed consultation in our study, busy occupation life was the second most common reason after negligence. Moreover, poor economic conditions are not only a single factor for the delay of TB diagnosis; a lack of time and convenience might also play an important role in this modern era. Non-enrollment in government health insurance was found to be a strong predictor of patient delay in our study. Similar results were found in a study conducted in Ghana [45]. Several factors contributed to non-enrollment in governments health insurance program, including lower household income, absence of chronic illness within the household, inappropriate benefit packages, cultural beliefs, and affordability issues [46,47]. The consistency in these findings highlights that socio-economic and cultural factors may play significant influence on health insurance enrollment decisions across different geographic and demographic contexts.

**Table 8. Factors associated with total delay among tuberculosis patients in the Kaski district.**

| Variables | Frequency (n) | Unadjusted | | Adjusted | |
|---|---|---|---|---|---|
| | | OR (95%CI) | p value | OR (95%CI) | p value |
| **Age** | | | | | |
| 18-40 years | 109 | Ref. | 0.432 | | |
| > 40 years | 85 | 1.37 (0.63–2.98) | | | |
| **Marital status** | | | | | |
| Married | 145 | Ref. | | | |
| Unmarried | 49 | 0.49 (0.22–1.10) | | | |
| **Education** | | | | | |
| Formal | 131 | Ref. | 0.013 | Ref. | 0.472 |
| Informal | 63 | **4.01 (1.34–11.99)**[*] | | 1.67 (0.41–6.75) | |
| **Occupation** | | | | | |
| Daily Wage Worker | 33 | 1.53 (0.50–4.69) | 0.460 | | |
| Others[a] | 161 | Ref. | | | |
| **Enrollment in Health Insurance Program** | | | | | |
| No | 148 | **3.16 (1.42–7.02)**[*] | 0.005 | 1.89 (0.67–5.28) | 0.227 |
| Yes | 46 | Ref. | | Ref. | |
| **Initially, visited health facility** | | | | | |
| Non-NTP providers | 89 | **5.81 (2.13–15.85)**[*] | 0.001 | **5.19 (1.53–17.61)**[*] | **0.008** |
| NTP providers | 105 | Ref. | | Ref. | |
| **Walking distance to HCF of first contact** | | | | | |
| > 30 Minutes | 52 | **6.70 (1.54–29.11)**[*] | 0.011 | 5.35 (0.87–32.86) | 0.070 |
| ≤ 30 Minute | 142 | Ref. | | Ref. | |
| **Smear status** | | | | | |
| Positive | 99 | Ref. | 0.795 | | |
| Negative | 96 | 1.11 (0.52–2.36 | | | |
| **Type of Tuberculosis** | | | | | |
| Extrapulmonary | 77 | 0.95 (0.44–2.07) | 0.906 | | |
| Pulmonary | 117 | Ref. | | | |
| **No. of HCF visited** | | | | | |
| 1 HCF | 59 | Ref. | <0.001 | Ref. | **0.024** |
| ≥ 2 HCFs | 135 | **5.26 (2.36–11.71)**[*] | | **3.86 (1.19–12.45)**[*] | |
| **No. of Times HCF visited** | | | | | |
| < 3 visits | 65 | Ref. | 0.001 | Ref. | 0.107 |
| ≥ 3 visits | 129 | **3.69 (1.68–8.07)**[*] | | 2.67 (0.81–8.63) | |
| **Smoking status** | | | | | |
| No | 77 | Ref. | 0.093 | | |
| Yes | 117 | 1.97 (0.90–4.14) | | | |
| **TB-related knowledge** | | | | | |
| Poor | 111 | **7.89 (3.10–20.53)**[*] | <0.001 | **5.40 (1.60–18.21)**[*] | **0.007** |
| Good | 83 | Ref. | | Ref. | |
| **Perceived Stigma** | | | | | |
| Low | 106 | Ref. | <0.001 | Ref. | **0.011** |
| High | 88 | **10.67 (3.13–36.44)**[*] | | **6.52 (1.54–27.57)**[*] | |

[*]Statistically significant at p < 0.05, Ref. = Reference value, HI = Health Insurance HCF = Healthcare Facility,

[a]Agriculture, Services, Business, House maker, Students, and Unemployed

## Factors associated with health-system delay

The findings of the study suggest various factors associated with health system delay. The use of tests other than gene-expert methods as initial diagnostic tools was found to be significantly associated with health system delays. In a study conducted in tuberculosis endemic settings such as southern Africa, it was found that the use of point-of-care GeneXpert was associated with fast treatment initiation, the start of treatment on the same day and treatment completion than traditional smear microscopy [44]. Additionally, other studies demonstrated that GeneXpert resulted in a reduction in the amount of time needed to start treatment after disease diagnosis [48]. Adequate availability of diagnostic equipment and supplies and human resources are essential for the proper functioning of peripheral health institutions. However, in a low-resource setting such as Nepal, sustaining these resources would also pose major challenges. The primary obstacles to the effective implementation of GeneXpert in Nepal were the absence of a cartridge supply, damaged module repair, GeneXpert machine maintenance and stock verification for prompt cartridge purchase [18]. Similarly, several studies conducted in Ethiopia, a low-resource country like Nepal, have shown that the absence of nearby health facilities offering TB diagnostic services such as the Xpert MTB/RIF assay and interruption of the supply of reagents (e.g., Cartridges) are the major barriers to case finding and treatment [18]. To solve these problems, robust supply chain management should be ensured by collaboration with suppliers, distributors and government agencies to ensure a proper supply of cartridges. Similarly, training should be provided to laboratory technicians and healthcare staff on how to maintain and repair GeneXpert machines in the laboratory.

Another significant factor associated with health system delay was the number of times a health care facility was visited. In Nepal, private service providers are more available and accessible than government facilities, and patients find private services trustworthy [12]. However, not all providers may offer TB diagnosis facilities. Thus, to identify all undiagnosed TB cases, the government needs to integrate private sector providers into the health system.

## Factors associated with total delay

In our study, the total delay was attributed to patient delay rather than to health system delay. This study concluded that having poor knowledge of TB was associated with unacceptable total delay in the treatment of TB. These findings are consistent with studies conducted in West Gojjam [49] and southern Ethiopia [50]. Poor TB-related knowledge was associated with unacceptable total delay. Not attending formal education was considered an important factor for unacceptable total delay in other settings because attending higher education might result in increased TB awareness and good treatment-seeking behavior.

TB stigma was found to be associated with total delay in TB diagnosis in this study. Another study conducted in Nepal revealed that stigma was considered a major barrier to treatment compliance [51]. The stigma and discrimination associated with TB may be due to the fear of perceived risk of infection, connection of TB with poverty and low caste, perceived links between TB and disreputable behaviors such as drinking alcohol, smoking tobacco and visiting sex workers, and perceptions that TB was a divine curse sent down to punish formerly unacceptable behavior [20]. This scenario highlights the urgent need for the government to reduce TB stigma in the community to achieve the goals and targets set in policies.

The TB service providers are both public and private in nature, some of which are recognized by NTP, while others are not [12]. Visiting non-NTP providers and healthcare facilities multiple times results in greater odds of total delay in diagnosis and treatment. This may be because informal providers might not be adequately trained to handle TB cases specific to the Nepalese context. Consequently, individuals visiting non-NTP providers upon experiencing TB symptoms may encounter prolonged delays. To mitigate this delay, efforts should focus on improving the health-seeking behavior of the public when symptoms of TB arise [33]. Additionally, non-NTP providers should be integrated into the NTP system, either by enabling them to provide services directly or by training them to refer patients to relevant facilities. This integration is crucial because evidence from similar socioeconomic and cultural settings in India indicates that involving private providers in TB services enhances TB and drug resistance TB diagnosis, notification, and treatment through early case

notification, referral and timely provision of diagnosis and treatment services [52]. The engagement of non-NTP private providers in the NTP system will also improve physical accessibility to the nearest healthcare service centers.

## Strengths and limitations of the study

The selection of samples and cases of TB were verified with the TB treatment card of the health management information system. The questionnaire was developed based on the World Health Organization's Multi-Country Study on Diagnosis and Treatment Delays in Tuberculosis [7].

The study may be subject to recall bias; however, the tool and the research were conducted in selected DOTS centers in the Kaski district only, thus limiting the generalizability of the results. The current study explored health system-related factors from the patients' perspective. Thus, further studies can be conducted to explore factors contributing to patient delay and health system delay from service providers and system perspectives.

## Conclusion

The study revealed that nearly three-fifths of the participants experienced delays in diagnosis and treatment, stemming from both patient and healthcare provider factors in Gandaki Province, western Nepal. These delays were primarily attributed to poor TB-related knowledge, patients seeking care from non-NTP service providers and visiting multiple healthcare facilities after symptom onset, and perceived stigma among the patients. Patient delay was predominant, and system delay was only assessed from the patient perspective. This highlights the necessity for future research to explore system delay from the viewpoints of health service providers and healthcare systems. In terms of total delay, perceived stigma and poor TB-related knowledge were identified as major barriers preventing individuals from seeking TB diagnosis and treatment services. Therefore, urgent action is required to implement targeted education campaigns aimed at raising TB awareness among public and TB patients, involving private sector and informal care providers in the NTP. Furthermore, it is imperative to develop and implement stigma reduction initiatives to address the issue of delay diagnosis and seeking care among TB patient in Nepal.

## Supporting information

**S1 Table. Number of participants in selected DOTS center.** This table presents the number of study participants from each of selected DOTS center, detailing patient enrollment across various facilities.
(XLSX)

**S1 File. Interview schedule.** This file contains the questions used for data collection in this study.
(DOCX)

**S1 Data. Final data.** This file contains the final data set used for analysis in this study.
(SAV)

## Author contributions

**Conceptualization:** Bikram Singh Dhami, Damaru Prasad Paneru, Dhirendra Nath.

**Data curation:** Bikram Singh Dhami, K.C. Aarati, Dhirendra Nath.

**Formal analysis:** Bikram Singh Dhami, Dhirendra Nath.

**Investigation:** Bikram Singh Dhami, Damaru Prasad Paneru, K.C. Aarati, Dhirendra Nath.

**Methodology:** Bikram Singh Dhami, Damaru Prasad Paneru, Dhirendra Nath.

**Project administration:** Bikram Singh Dhami, Sagar Parajuli, K.C. Aarati, Dhirendra Nath.

**Resources:** Bikram Singh Dhami, Sagar Parajuli, K.C. Aarati, Dhirendra Nath.

**Software:** Bikram Singh Dhami, Damaru Prasad Paneru, K.C. Aarati, Dhirendra Nath.

**Supervision:** Damaru Prasad Paneru, Dhirendra Nath.

**Validation:** Damaru Prasad Paneru, Dhirendra Nath.

**Visualization:** Dhirendra Nath.

**Writing – original draft:** Bikram Singh Dhami, Sagar Parajuli, Dhirendra Nath.

**Writing – review & editing:** Bikram Singh Dhami, Damaru Prasad Paneru, Sagar Parajuli, K.C. Aarati, Dhirendra Nath.

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
