## [Decision Letter · Decision Letter 0]

9 Oct 2024

PGPH-D-24-01716

Delays in Diagnosis and Treatment among Tuberculosis Patients in Gandaki, Nepal

Dear Dr. Dhami and Colleagues,

Thank you for submitting your manuscript to PLOS Global Public Health. After careful consideration, we feel that it has merit but does not fully meet PLOS Global Public Health’s publication criteria as it currently stands. Therefore, we invite you to submit a revised version of the manuscript that addresses the points raised during the review process.

Thank you, also, for the opportunity to read and consider your paper for publication. As you'll likely see below both of your peer reviewers found considerable scientific merit in the article and view it as an important contribution to conversations around delays in TB care and treatment both in South Asia and globally. From my end I feel that meets PLoS Global Public Health's requirements for inclusion in the journal. They both suggested that the article might, however, benefit from some minor revisions and I agree with them. These revisions center on adding a bit more precision to your language, slightly modifying the way that the paper talks about delay, and sharing important contextual information which may be of benefit for the reader. I agree with them here too. In particular reviewer one has invited you to share more information concerning the steps you took to ensure the rigor and validity of your data, namely validating tools in Nepali and the steps taken by data collectors to ensure accurate representation both in documentation and in accounts collected. These seem central ways to strengthen the paper and I hope you'll follow up on the reviewer's suggestions. Both have sent diligent line based comments which I hope you will read carefully. To my eye this suggests that there is no need for a restructuring of the article as submitted but a careful round of editing for clarity and being sure that the reader has the tools to engage with your insights. Similarly I find the reviewer comments suggesting revisions to the tables and finding out where the missing figure has gone particularly useful and I hope you'll take them. Last, reviewer two has suggested that you slightly revise the conclusion to better represent your what your results do suggest. Though education initiatives are important, reviewer two seems to suggest that your data actually calls for more or different forms of intervention. I, along with reviewer two, invite you to share them with readers in the conclusion. Also, remember as you revise that PLoS Global Public Health has a global audience and is read by people with a great many ways of using academic English. It will help your paper reach more audiences if you take steps to write in short clear sentences that stick rather closely to standard forms of English. Overall the paper already does this well but as you go through this round of minor revisions I encourage you to take any opportunity you find to use clear, plain language. I am grateful for the opportunity to engage your paper and certainly hope you will be willing to lightly revise the manuscript along these two reviewers' suggestion. I think it will help make an already strong paper an excellent one.

We look forward to receiving your revised manuscript.

Kind regards,

Andrew James McDowell

Academic Editor

Journal Requirements:

1. Please provide a complete Data Availability Statement in the submission form, ensuring you include all necessary access information or a reason for why you are unable to make your data freely accessible. If your research concerns only data provided within your submission, please write "All data are in the manuscript and/or supporting information files" as your Data Availability Statement.

Reviewers' comments:

Reviewer's Responses to Questions

**Comments to the Author**

1. Does this manuscript meet PLOS Global Public Health’s publication criteria ? Is the manuscript technically sound, and do the data support the conclusions? The manuscript must describe methodologically and ethically rigorous research with conclusions that are appropriately drawn based on the data presented.

Reviewer #1: Yes

Reviewer #2: Yes

2. Has the statistical analysis been performed appropriately and rigorously?

Reviewer #1: Yes

Reviewer #2: Yes

3. Have the authors made all data underlying the findings in their manuscript fully available (please refer to the Data Availability Statement at the start of the manuscript PDF file)?

Reviewer #1: Yes

Reviewer #2: Yes

4. Is the manuscript presented in an intelligible fashion and written in standard English?

Reviewer #1: Yes

Reviewer #2: Yes

5. Review Comments to the Author

Reviewer #1: It seems you have tried to incorporate all the things from your thesis report. I suggest you make it readable.

Title: This study title and objectives are to identify factors that delays in diagnosis and treatment but your results incorporate patients delay and health system delay. You have a figure 1 that shows diagnosis delays and treatment delays as others delay as well. Also, you have total delays which includes all of it. How do you correlate? It’s a bit confusing. Please provide detail justification.

Line 123: Only DOTS can be used after illustrating its full form in previous lines.

Line 126: Why did this study excluded the participants whose treatment failed? Justify.

Line 130: what was the type of delay from previous study used? Was it delay in seeking care or receiving treatment?

Line 134: Provide the number of participants from 10 DOTS centers in supplementary file.

Line 153: You have used extensive literature review. Was it that extensive? You can remove the word extensive.

Line 155: Were the tools used was validated in Nepali language? If yes cite it. If not, how do you maintain the validity of the tools in Nepali language?

Line 179: How did your data collectors (public health students) maintain the credibility of responses for clinical characteristics? How was it done? Justify.

Line 187: Cite the reference. Was this scoring validated from standard tools?

Line 192: Again, cite the relevant reference for the Likert scale used.

Line 247: Is it P value less than or equals 0.05? Please check. If yes, please explain?

Line 249: Report the highest VIF value from your data.

Line 302: The table is large. Split the table and use different table headings. If some characteristics are multiple response, use notation for it.

Line 308: Use only DOTS.

Line 310: Table 3. Note as multiple response where necessary. Check all tables again.

Line 323: Figures are missing. Check it.

Line 282-385: Cite proper references.

Reviewer #2: 1. Abstract: The first sentence in the background should not directly link to increased drug resistance. It may lead to that in some cases such as where patients undergo incomplete treatments. Mere delays will not lead to increased resistance. It needs to be worded carefully.

2. Line 15: multi-country

3. Patient and health system delays need to be calculated separately. Even after reaching the health system (first point as private), further delays in diagnosis/treatment may be due to patients’ behaviour. It seems there is an overlap. Patient related delays are more likely linked to socio-economic status, stigma, illiteracy etc. as seen from the results, whereas health system delays are more likely related to system related issues. They both need different kinds of interventions. The multivariate analysis looks more appropriate.

4. Conclusion: Needs to be more specific than just education/training. Factors that caused significant delays, should be focused while suggesting the intervention measures. Private providers’ lack of suspicion of TB is one of the reasons for delayed diagnosis and treatment.

5. Line 89 should be GeneXpert or Xpert assay.

6. Lines 98-99: The rationale is unclear. Can authors make it explicit?

7. Lines 108-112: Reference needs to be provided. Authors can provide the reference in the first sentence and mention about the same report---etc.

8. Line 273: 35.6% cannot be the most. Better to say around a third

9. Lines 295-298: It is commonly observed in S. Asia that majority patients approach private health facility and over two third patients reported visits to >3 providers before getting TB diagnosis. The same experience was reported in India (Re. Atre et al. 2022 AJRCCM paper on pathways to TB and MDR-TB care).

10. Can authors elaborate reasons for patients’ non-enrolment in Govt insurance program?

6. PLOS authors have the option to publish the peer review history of their article (what does this mean? ). If published, this will include your full peer review and any attached files.

**Do you want your identity to be public for this peer review?** For information about this choice, including consent withdrawal, please see our Privacy Policy .

Reviewer #1: **Yes: ** Prabin Karki

Reviewer #2: **Yes: ** Sachin Atre

---

## [Decision Letter · Decision Letter 1]

2 May 2025

Patient and Health System Delays in the Diagnosis and Treatment of Tuberculosis in Gandaki, Nepal

PGPH-D-24-01716R1

Dear Dr. Dhami,

We are pleased to inform you that your manuscript 'Patient and Health System Delays in the Diagnosis and Treatment of Tuberculosis in Gandaki, Nepal' has been provisionally accepted for publication in PLOS Global Public Health.

Best regards,

Shifa S. Habib

Academic Editor

Reviewer Comments (if any, and for reference):

Reviewer's Responses to Questions

**Comments to the Author**

1. If the authors have adequately addressed your comments raised in a previous round of review and you feel that this manuscript is now acceptable for publication, you may indicate that here to bypass the “Comments to the Author” section, enter your conflict of interest statement in the “Confidential to Editor” section, and submit your "Accept" recommendation.

Reviewer #2: All comments have been addressed

Reviewer #3: All comments have been addressed

Reviewer #4: (No Response)

2. Does this manuscript meet PLOS Global Public Health’s publication criteria ? Is the manuscript technically sound, and do the data support the conclusions? The manuscript must describe methodologically and ethically rigorous research with conclusions that are appropriately drawn based on the data presented.

Reviewer #2: Yes

Reviewer #3: Yes

Reviewer #4: Yes

3. Has the statistical analysis been performed appropriately and rigorously?

Reviewer #2: I don't know

Reviewer #3: Yes

Reviewer #4: Yes

4. Have the authors made all data underlying the findings in their manuscript fully available (please refer to the Data Availability Statement at the start of the manuscript PDF file)?

Reviewer #2: Yes

Reviewer #3: Yes

Reviewer #4: Yes

5. Is the manuscript presented in an intelligible fashion and written in standard English?

Reviewer #2: Yes

Reviewer #3: Yes

Reviewer #4: Yes

6. Review Comments to the Author

Reviewer #2: All my comments have been addressed.

Reviewer #3: (No Response)

Reviewer #4: (No Response)

7. PLOS authors have the option to publish the peer review history of their article (what does this mean? ). If published, this will include your full peer review and any attached files.

**Do you want your identity to be public for this peer review?** For information about this choice, including consent withdrawal, please see our Privacy Policy .
